# A Comprehensive Assessment of Bioactive Metabolites, Antioxidant and Antiproliferative Activities of *Cyclocarya paliurus* (Batal.) Iljinskaja Leaves

**Mingming Zhou [1], Pei Chen [1], Yuan Lin [1], Shengzuo Fang [1,2],* and Xulan Shang [1,2]**

1   College of Forestry, Nanjing Forestry University, Nanjing 210037, China
2   Co-Innovation Center for Sustainable Forestry in Southern China, Nanjing Forestry University, Nanjing 210037, China
*   Correspondence: fangsz@njfu.edu.cn or fangsz@njfu.com.cn; Tel./Fax: +86-258-542-7797

**Abstract:** *Cyclocarya paliurus* (Batal.) Iljinskaja is an indigenous and multifunction tree species in China, but it is mainly used in pharmaceutical and nutraceutical ingredients. To make a comprehensive evaluation on its bioactive metabolites, antioxidant and antitumor potentials of *C. paliurus* leaves, the leaf samples were collected from 15 geographic locations (natural populations) throughout its distribution areas. High-performance liquid chromatography (HPLC) and colorimetric methods were used to detect the contents of bioactive metabolites. The antioxidant activity was evaluated by 2,2′-diphenyl-1-picrylhydrazyl (DPPH), 2,2-azino-bis (3-ethylbenzothiazoline-6-sulfonic acid) (ABTS) and reducing power assays. The antiproliferative activity on different cancer cell types was evaluated by 3-(4,5-dimethylthiazol-2-yl)-2,5-diphenyltetrazolium bromide (MTT) assay. Contents of bioactive metabolites, and antioxidant and antiproliferative activities in the extracts were significantly affected by solvent and population. In most cases, the contents of flavonoids and triterpenoids, and the antioxidant and antiproliferative activities in the ethanol extracts were higher than the water extracts. The best scavenging capacity of DPPH ($IC_{50}$ = 0.34 mg/mL) and ABTS ($IC_{50}$ = 0.50 mg/mL) radical occurred in the ethanol extracts of S15 and S7 population respectively, while the strongest reducing power ($EC_{50}$ = 0.71 mg/mL) was achieved in the ethanol extracts of S14 population. The antiproliferation effects of *C. paliurus* extracts on cancer cells varied with different cell types. The HeLa cell was the most sensitive to *C. paliurus* extracts, and their $IC_{50}$ values of the ethanol extracts varied from 0.13 to 0.42 mg/mL among *C. paliurus* populations. Redundancy analysis showed that total polyphenol had the greatest contribution to the antioxidant activity, but total flavonoid was mostly responsible for the antiproliferation effects. These results would provide important scientific evidences not only for developing *C. paliurus* as a potent antioxidant and antitumor reagent, but also for obtaining the higher yield of bioactive compounds in the *C. paliurus* plantation.

**Keywords:** *Cyclocarya paliurus*; flavonoid; phenolics; triterpenoid; solvent; natural population

---

## 1. Introduction

Cancer, a disease characterized by abnormal cells that divide and invade other tissues and organs uncontrollably, is one of the leading causes of unnatural death in the world [1]. To date, there are more than 100 types of cancer, but the common cancers are lung, breast, colorectal and stomach [2]. Surgery, chemotherapy and radiation are commonly used for cancer treatment along with side effects [3]. Over the past decades, natural products have attracted a great deal of attention due to their utilization of cancer prevention and treatment. Over 60% of anticancer drugs are derived from natural products [4].

Most anticancer drugs, such as taxol, camptothecin, vinblastine, vincristine and podophyllotoxin, originate from plant resources [4,5]. The bioactive metabolites of plant extracts, such as phenolics, alkaloids and terpenoids, are considered as potential anticancer reagents [4,6,7].

Oxidative damage plays an important role in the occurrence of various diseases including cancer. Reactive oxygen species (ROS) could result in oxidative damage of cell components such as lipids, protein and DNA, and thereby induce cancer cell formation, transformation and cancer-promoting signaling molecules [8]. Antioxidants can protect cells from oxidative damage though neutralizing ROS. In recent years, many studies have demonstrated that the intake of plant products that are rich in antioxidant phytochemicals can reduce the incidence of many diseases including cancer [9,10]. Therefore, it is important to seek natural antioxidants with anticancer potentials.

*Cyclocarya paliurus* (Batal.) Iljinskaja, an indigenous and multifunction tree species in China, is mainly distributed in highlands of subtropical areas. Its leaves have been made into nutraceutical tea beverage for a long time in folk tradition and served as a formula of traditional Chinese medicine [11]. Moreover, the leaves of *C. paliurus* have been listed as new food raw material by National Health and Family Planning Commission of China since 2013 [12]. Pharmacological studies on *C. paliurus* indicated that its extracts had a variety of biological activities such as antidiabetic, antioxidant and antibacterial activities, while less attention was paid to the antiproliferation effects of *C. paliurus* on different cancer cells. Previous studies had revealed that the abundant phytochemicals in the extracts of *C. paliurus* leaves are responsible for these bioactivities. For instance, hypolipidemic and hypoglycemic effects of *C. paliurus* were ascribed to the flavonoids and triterpenoids in its extracts respectively [13], while polysaccharides isolated from the water extracts of *C. paliurus* exhibited hypolipidemic effects and antiproliferation effects on HeLa cell through cell-cycle arrest in the S phase [14,15]. Additionally, *C. paliurus* polysaccharides showed significant scavenging capacity on 2,2'-diphenyl-1-picrylhydrazyl (DPPH) radicals at $IC_{50}$ value of 52.3 µg/mL [16]. Total polyphenol among phenolics and polysaccharide exhibited the greatest contribution to antioxidant activity in aqueous extracts of *C. paliurus* leaves [17]. Quercetin and kaempferol glycosides of phenolic compounds were key antioxidant components in ethanol extracts of *C. paliurus* leaves [18]. However, there was less information available on the relationship between antioxidant activity and triterpenoid in *C. paliurus* extracts. Recently, there have been numerous studies on plants products in order to seek key components with antioxidant and antitumor properties. Although polysaccharides, phenolics and triterpenoids have antiproliferation effects on cancer cells and antioxidant activity in some plants [15,19], the main bioactive constituents with antioxidant and antiproliferative activities remain unclear in leaf extracts of *C. paliurus*. Polysaccharide content and antioxidant activity of water extracts of *C. paliurus* leaves collected from different populations were investigated in previous studies [14,17]. Moreover, no polysaccharide was found in the ethanol extracts [13]. In this context, it is crucial to study the bioactive components belonging to phenolics or triterpenoids for antioxidant and antitumor potentials in *C. paliurus* leaves.

Geographical origin has a pivotal impact on phytochemicals and biological activities. The contents of polysaccharide, phenolics, triterpenoid and their antioxidant, hypolipidemic and hypoglycemic effects in *C. paliurus* leaves have been reported to vary with different geographical locations [13,17,18,20]. Phytochemical contents, antioxidant and anticancer activities in plant extracts also vary with extraction solvents such as water, aqueous mixtures of ethanol, ethanol and methanol [21,22]. For example, antiproliferative activity on cancer cells was found in both ethanol and aqueous extracts of *Ficus beecheyana* and *Saururus chinensis* root, however the ethanol extracts showed better antioxidant activity [23,24]. The phenolic compounds, anticancer and antioxidant activities of indigo plants also varied with the extraction solvents [25]. In *C. paliurus* leaves, the chemical composition and antidiabetic effects in vivo have been evaluated in both aqueous and ethanol extracts from five locations [13]. However, to our best knowledge, there was no comprehensive assessment on antioxidant and antiproliferative activities of *C. paliurus* leaves considering both solvents and geographical origins.

In this framework, the contents of phenolics and triterpenoids, antioxidant and antiproliferation effects of both ethanol and water extracts were investigated in *C. paliurus* leaves collected from different geographical locations. Our aims were to detect which solvent was more efficient for antioxidant and antitumor potentials, seek for the components that were responsible for antioxidant and antiproliferative activities in *C. paliurus* leaves, and screen out the superior natural populations with targeted bioactivity. The results will not only offer a theoretical basis for obtaining the higher yield of bioactive compounds from *C. paliurus* leaves, but also provide a scientific basis for developing *C. paliurus* as a potential antioxidant and antitumor reagent.

## 2. Materials and Methods

### 2.1. Plant Material

The leaf samples of *C. paliurus* were collected from 15 geographical locations of its natural distribution areas in current study (Figure 1). At each location, the mature leaves from 6 to 30 dominant or co-dominant trees were sampled. And then the leaves collected from trees per location were mixed as a population sample. The sample collection and pre-treatment followed the method of Liu et al. [20]. Finally, the *C. paliurus* leaves were dried to constant weight at 70 °C and then grinded into powder. All samples were stored at room temperature prior to analysis.

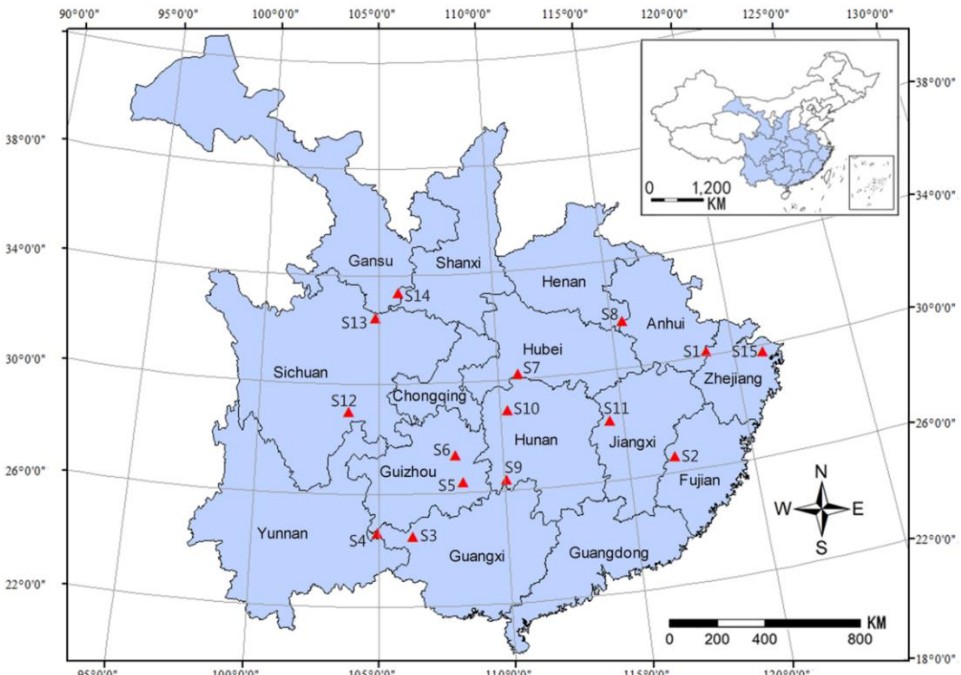

**Figure 1.** A map showing natural distribution across 15 province or municipalities in China and geographical locations of *C. paliurus* populations sampled in this study (red triangle). The populations codes were as follows: S1 (Jixi) from Anhui province; S2 (Mingxi) from Fujian province; S3 (Baise) and S4 (Jinzhongshan) from Guangxi province; S5 (Jianhe) and S6 (Shiqian) from Guizhou province; S7 (Wufeng) from Hubei province; S8 (Shangcheng) from Henan province; S9 (Suining) and S10 (Yongshun) from Hunan province; S11 (Xiushui) from Jiangxi province; S12 (Muchuan) and S13 (Qingchuan) from Sichuan province; S14 (Lueyang) from Shanxi province; S15 (Fenghua) from Zhejiang province.

### 2.2. Preparation of Extracts

Each 3.0 g leaf sample was extracted with 30 mL of 70% ethanol and water for 1 h at 90 °C, respectively. And then they were subjected to ultrasound for 45 min. The mixtures were centrifuged at 10,000 rpm for 10 min and the supernatants were retained. The obtained extracts were separated on C18 solid phase extraction column for determination. For high-performance liquid chromatography

(HPLC) analysis, the extracts were filtered through a 0.22 μm syringe filter. Each sample was analyzed in triplicate.

### 2.3. Content Determination of Bioactive Metabolites

Total polyphenol content was determined by the Folin-Ciocalteu (FC) colorimetric method as described by Xie et al. [26], while total flavonoid content was determined using the aluminum trichloride colorimetric method as described by Li et al. [27], and total triterpenoid content was measured by the colorimetric method as described by Fan and He [28]. HPLC analysis was conducted to detect the phenolic acid, flavonoid and triterpenoid content by following the method of Cao et al. [11]. The chromatograms of the representative sample and the 16 mixed standards were shown in Supplementary Figure S1.

### 2.4. Antioxidant Assay

Six concentration gradients of ethanol extracts (1 mL, 0.2–1.2 mg/mL) and water extracts (1 mL, 0.4–2.4 mg/mL) were designed by equal interval to detect the antioxidant capacities. Sample solvent (70% ethanol or distilled water) was used as the control. The scavenging capacity of DPPH and 2,2-azino-bis (3-ethylbenzothiazoline-6-sulfonic acid) (ABTS )radicals, and the reducing power were determined and analyzed by the methods described by Zhou et al. [17]. The percent inhibition of radical was calculated as $((A_{control} - A_{sample})/A_{control}) \times 100\%$, where $A_{control}$ and $A_{sample}$ are the absorbances of the control and sample, respectively.

### 2.5. Anticancer Assay

#### 2.5.1. Cell Culture

All cell lines used in this study originated from human bodies. Renal cell line (HEK293) was used as human normal cells, whereas the colon cancer cell line (HCT-116), cervical cancer cell line (HeLa), liver cancer cell line (HepG2), breast cancer cell line (MCF-7), lung cancer cell line (A549), and pancreatic cancer cell line (PANC-1) were used as the human cancer cell lines in this study. Different mediums were used for the incubation of cell lines. HEK-293 and PANC-1 were incubated by DMEM (Dulbeco's Modified Eagle's Medium) after adding 10% FBS (fetal bovine serum). MCF-7 and A549 were incubated by RPMI1640 (Roswell Park Memorial Institute medium) after adding 10% FBS. HeLa and HepG2 were incubated by MEM (Minimum Essential Medium) after adding 10% FBS. HCT-116 was incubated by McCoy's 5A Medium after adding 10% FBS. Cell lines were maintained in mediums at 37 °C and 5% $CO_2$ incubator.

#### 2.5.2. MTT Assay

Inhibition effects on the proliferation of cancer cells were determined using MTT assay. The cells were adjusted at a concentration of $4 \times 10^4$ cells/mL, and then 100 μL cell suspension was added to each well in 96 well microplates and incubated at 37 °C and 5% $CO_2$ for 24 h. Each of 100 μL extracts of different concentrations (100, 200, 400, 800, and 1600 μg/mL) were added to 96 well microplates. Then, cells were incubated for 72 at 37 °C and 5% $CO_2$. After adding 20 μL MTT (5 mg/mL) to per wells, the cells were incubated for 4 at 37 °C and 5% $CO_2$. Then culture media was removed and 150 μL dimethyl sulfoxide (DMSO) was added to per well, and then stirred for 10 min. Absorbance was measured at 490 nm with microplate reader. Sample solvent (70% ethanol or distilled water) was used as the control. The inhibition percentage of cell proliferation was calculated by the following equation: $((A_{control} - A_{sample})/A_{control}) \times 100\%$, where $A_{control}$ and $A_{sample}$ are the absorbance of the control and sample, respectively.

### 2.6. Statistical Analysis

All values were expressed as means ± standard deviations (SD). One-way analysis of variance (ANOVA) was used for test the variation of phytochemicals, antioxidant and anticancer potentials among *C. paliurus* populations, followed by Turkey range test with $p = 0.05$. General linear model (GLM) analysis was performed to detect the effects of solvents and geographical origin on phytochemical contents, antioxidant and anticancer properties. The above analyses were performed by SPSS 20.0 software (SPSS Inc., Chicago, IL, USA). Redundancy analysis (RDA) was carried out to detect the correlation between phytochemicals and bioactivity by Canoco 4.5 software (Wageningen UR, Wageningen, The Netherlands).

## 3. Results and Discussion

### 3.1. Effects of Solvent and Geographical Origin on Bioactive Metabolites

The contents of bioactive metabolites in *C. paliurus* leaves were significantly affected by solvent and population. Based on the F-values, solvent had larger effects on bioactive metabolites except for 3-*O*-caffeoyluinic acid and quercetin-3-*O*-glucuronide (Supplementary Table S1). Additionally, there were significant interaction effects between solvent and population on bioactive metabolites (Supplementary Table S1). The amount of total polyphenol, total flavonoid and total triterpenoid varied from 4.85 to 52.67 mg/g, 2.81 to 20.72 mg/g and 2.61 to 50.67 mg/g, respectively. The highest contents were found in ethanol extracts of S7 population, while the lowest contents were recorded in water extracts of S9 population (Figure 2). Furthermore, the HPLC analysis showed that in most cases, the water extracts exhibited higher 3-*O*-caffeoyluinic acid (from 0.10 to 2.29 mg/g) and 4-*O*-caffeoyluinic acid (from 0.03 to 0.48 mg/g), and the lowest values appeared in S9 population (Table 1). However, the ethanol solvent was more effective for the extraction of flavonoids and triterpenoids in most *C. paliurus* populations. Quercetin-3-*O*-glucuronide (from 0.14 to 3.71 mg/g) was found to be the dominated flavonoid, followed by kaempferol-3-*O*-rhamnoside (from 0.16 to 2.73 mg/g) and kaempferol-3-*O*-glucuronide (from 0.14 to 1.77 mg/g) (Table 2). Arjunolic acid ranged from 0.23 to 4.60 mg/g was the leading component of triterpenoids in the ethanol extracts. In addition, S11 population was superior to all other populations with respect to the individual triterpenoid (Table 3).

**Table 1.** The content (mg/g) of phenolic acid in ethanol and water extracts of *C. paliurus* leaves.

| Populations | Ethanol | | | Water | | |
|---|---|---|---|---|---|---|
| | P1 | P2 | P3 | P1 | P2 | P3 |
| S1 | 1.23 ± 0.01[c] | 0.24 ± 0.00[b] | 0.33 ± 0.02[b] | 1.43 ± 0.05[bc] | 0.42 ± 0.03[a] | 0.03 ± 0.00[c] |
| S2 | 0.14 ± 0.02[i] | 0.05 ± 0.01[fg] | nd | 0.22 ± 0.02[hi] | 0.09 ± 0.01[de] | nd |
| S3 | 1.65 ± 0.07[a] | 0.11 ± 0.00[e] | 0.19 ± 0.01[d] | 2.29 ± 0.06[a] | 0.29 ± 0.01[b] | nd |
| S4 | 0.15 ± 0.01[i] | 0.04 ± 0.01[g] | nd | 0.32 ± 0.04[hi] | 0.06 ± 0.01[de] | nd |
| S5 | 0.84 ± 0.03[e] | 0.16 ± 0.01[d] | 0.19 ± 0.00[d] | 0.92 ± 0.01[ef] | 0.25 ± 0.00[b] | nd |
| S6 | 0.68 ± 0.01[f] | 0.15 ± 0.00[d] | 0.18 ± 0.00[d] | 1.12 ± 0.01[de] | 0.27 ± 0.01[b] | 0.01 ± 0.00[d] |
| S7 | 1.07 ± 0.02[d] | 0.31 ± 0.01[a] | 0.26 ± 0.01[c] | 1.24 ± 0.18[cd] | 0.47 ± 0.07[a] | 0.04 ± 0.00[a] |
| S8 | 0.49 ± 0.03[g] | 0.11 ± 0.01[e] | 0.05 ± 0.01[e] | 0.66 ± 0.01[fg] | 0.26 ± 0.01[b] | nd |
| S9 | 0.26 ± 0.03[h] | 0.06 ± 0.00[f] | nd | 0.10 ± 0.01[i] | 0.03 ± 0.00[e] | nd |
| S10 | 0.23 ± 0.02[hi] | 0.05 ± 0.00[fg] | nd | 0.76 ± 0.01[f] | 0.20 ± 0.00[bc] | nd |
| S11 | 0.67 ± 0.06[f] | 0.19 ± 0.02[c] | 0.20 ± 0.03[d] | 0.41 ± 0.01[gh] | 0.10 ± 0.01[de] | nd |
| S12 | 1.44 ± 0.01[b] | 0.21 ± 0.01[bc] | 0.22 ± 0.01[cd] | 2.18 ± 0.04[a] | 0.43 ± 0.02[a] | nd |
| S13 | 0.59 ± 0.00[f] | 0.09 ± 0.00[e] | 0.19 ± 0.01[d] | 0.43 ± 0.01[gh] | 0.13 ± 0.00[cd] | nd |
| S14 | 1.61 ± 0.02[a] | 0.33 ± 0.01[a] | 0.47 ± 0.04[a] | 1.73 ± 0.33[b] | 0.48 ± 0.08[a] | 0.04 ± 0.00[b] |
| S15 | 0.15 ± 0.03[i] | 0.06 ± 0.01[fg] | nd | 0.43 ± 0.05[gh] | 0.20 ± 0.02[bc] | nd |

P1: 3-*O*-caffeoyluinic acid; P2: 4-*O*-caffeoyluinic acid; P3: 4,5-di-*O*-caffeoyluinic acid; nd: not detected. Values within different superscripts are different in the same column at 0.05 levels.

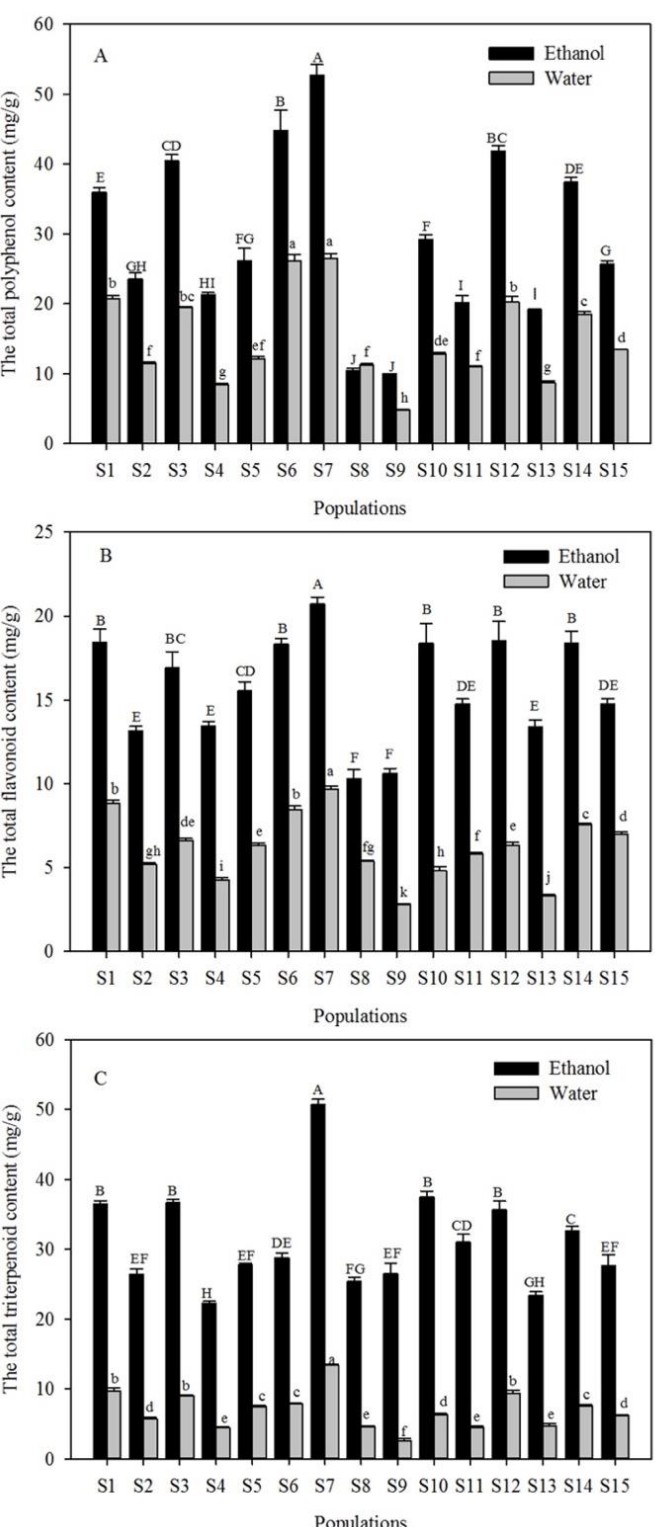

**Figure 2.** The contents of total polyphenol (**A**), total flavonoid (**B**) and total triterpenoid (**C**) in ethanol and water extracts of *C. paliurus* leaves. Different capital and small letters indicated significant differences in the ethanol and water extracts at 0.05 levels, respectively.

**Table 2.** The content (mg/g) of individual flavonoid in ethanol and water extracts of *C. paliurus* leaves.

| Populations | Ethanol | | | | | | | Water | | | | | | |
|---|---|---|---|---|---|---|---|---|---|---|---|---|---|---|
| | F1 | F2 | F3 | F4 | F5 | F6 | F7 | F1 | F2 | F3 | F4 | F5 | F6 | F7 |
| S1 | 3.71 ± 0.02$^a$ | 0.76 ± 0.00$^a$ | 0.23 ± 0.00$^{fg}$ | 1.55 ± 0.01$^{abc}$ | 0.11 ± 0.00$^{gh}$ | 0.33 ± 0.00$^a$ | 2.73 ± 0.02$^a$ | 2.30 ± 0.03$^a$ | 0.24 ± 0.02$^b$ | 0.08 ± 0.00$^b$ | 0.95 ± 0.08$^{bc}$ | 0.07 ± 0.01$^{bc}$ | 0.13 ± 0.01$^a$ | 0.66 ± 0.27$^a$ |
| S2 | 0.20 ± 0.04$^g$ | 0.05 ± 0.00$^{fg}$ | 0.02 ± 0.00$^h$ | 0.15 ± 0.03$^h$ | 0.06 ± 0.00$^h$ | 0.02 ± 0.00$^f$ | 0.83 ± 0.06$^f$ | 0.33 ± 0.05$^h$ | 0.05 ± 0.00$^{gh}$ | 0.01 ± 0.00$^e$ | 0.25 ± 0.03$^{fg}$ | nd | 0.02 ± 0.00$^{gh}$ | 0.07 ± 0.01$^{hi}$ |
| S3 | 2.26 ± 0.06$^b$ | 0.81 ± 0.03$^a$ | 0.68 ± 0.02$^b$ | 1.06 ± 0.04$^e$ | 0.32 ± 0.01$^d$ | 0.23 ± 0.01$^b$ | 1.69 ± 0.05$^b$ | 1.40 ± 0.04$^c$ | 0.24 ± 0.01$^b$ | 0.06 ± 0.00$^{bc}$ | 0.64 ± 0.02$^e$ | 0.06 ± 0.00$^{bc}$ | 0.08 ± 0.00$^{bc}$ | 0.44 ± 0.01$^c$ |
| S4 | 0.19 ± 0.01$^g$ | 0.04 ± 0.00$^g$ | 0.01 ± 0.00$^h$ | 0.20 ± 0.08$^{gh}$ | 0.06 ± 0.01$^h$ | 0.03 ± 0.00$^f$ | 0.16 ± 0.01$^h$ | 0.35 ± 0.06$^h$ | 0.04 ± 0.00$^{hi}$ | 0.01 ± 0.00$^e$ | 0.24 ± 0.03$^{fgh}$ | nd | 0.03 ± 0.01$^{e-h}$ | 0.11 ± 0.02$^{gh}$ |
| S5 | 1.22 ± 0.02$^e$ | 0.55 ± 0.01$^c$ | 0.28 ± 0.01$^f$ | 1.30 ± 0.02$^{cde}$ | 0.20 ± 0.00$^{ef}$ | 0.14 ± 0.00$^c$ | 1.07 ± 0.02$^e$ | 0.65 ± 0.00$^{ef}$ | 0.12 ± 0.00$^d$ | 0.02 ± 0.00$^e$ | 0.70 ± 0.00$^{de}$ | nd | 0.04 ± 0.00$^{def}$ | 0.21 ± 0.00$^e$ |
| S6 | 2.15 ± 0.04$^b$ | 0.63 ± 0.01$^b$ | 0.36 ± 0.01$^e$ | 1.51 ± 0.03$^{bc}$ | 0.15 ± 0.00$^{fg}$ | 0.07 ± 0.00$^{de}$ | 0.45 ± 0.01$^g$ | 1.69 ± 0.02$^b$ | 0.28 ± 0.00$^a$ | 0.15 ± 0.00$^a$ | 1.08 ± 0.01$^{ab}$ | 0.07 ± 0.00$^{bc}$ | 0.04 ± 0.00$^{de}$ | 0.15 ± 0.00$^{fg}$ |
| S7 | 2.01 ± 0.07$^{bc}$ | 0.34 ± 0.01$^e$ | 0.16 ± 0.00$^g$ | 1.37 ± 0.06$^{cd}$ | 0.14 ± 0.02$^{fgh}$ | 0.17 ± 0.01$^c$ | 1.47 ± 0.06$^c$ | 1.65 ± 0.04$^b$ | 0.18 ± 0.02$^c$ | 0.02 ± 0.00$^{de}$ | 1.17 ± 0.02$^a$ | 0.06 ± 0.00$^{bc}$ | 0.09 ± 0.01$^b$ | 0.56 ± 0.00$^b$ |
| S8 | 0.56 ± 0.04$^f$ | 0.13 ± 0.01$^f$ | 0.05 ± 0.00$^h$ | 0.78 ± 0.06$^f$ | 0.09 ± 0.00$^{gh}$ | 0.09 ± 0.01$^d$ | 0.80 ± 0.06$^f$ | 0.42 ± 0.01$^{gh}$ | 0.09 ± 0.00$^{ef}$ | 0.03 ± 0.00$^{de}$ | 0.62 ± 0.01$^e$ | 0.06 ± 0.00$^{bc}$ | 0.04 ± 0.00$^{d-g}$ | 0.12 ± 0.00$^{gh}$ |
| S9 | 0.21 ± 0.01$^g$ | 0.06 ± 0.00$^{fg}$ | 0.02 ± 0.00$^h$ | 0.30 ± 0.01$^{gh}$ | 0.06 ± 0.01$^h$ | 0.04 ± 0.00$^{ef}$ | 0.24 ± 0.03$^h$ | 0.07 ± 0.00$^i$ | 0.02 ± 0.00$^i$ | 0.01 ± 0.00$^e$ | 0.08 ± 0.01$^h$ | nd | 0.02 ± 0.00$^h$ | 0.03 ± 0.00$^i$ |
| S10 | 0.20 ± 0.03$^g$ | 0.05 ± 0.01$^g$ | 0.03 ± 0.01$^h$ | 0.14 ± 0.02$^h$ | 0.07 ± 0.00$^{gh}$ | 0.03 ± 0.00$^f$ | 0.16 ± 0.03$^h$ | 0.55 ± 0.01$^{fg}$ | 0.05 ± 0.00$^{gh}$ | 0.01 ± 0.00$^e$ | 0.32 ± 0.02$^f$ | 0.09 ± 0.00$^b$ | 0.02 ± 0.00$^{gh}$ | 0.18 ± 0.00$^{ef}$ |
| S11 | 1.44 ± 0.16$^{de}$ | 0.63 ± 0.07$^{bc}$ | 0.40 ± 0.05$^{de}$ | 1.74 ± 0.19$^{ab}$ | 0.25 ± 0.02$^{de}$ | 0.22 ± 0.02$^b$ | 1.26 ± 0.14$^d$ | 0.46 ± 0.07$^{gh}$ | 0.08 ± 0.01$^{ef}$ | 0.01 ± 0.00$^e$ | 0.55 ± 0.09$^e$ | 0.05 ± 0.00$^c$ | 0.04 ± 0.01$^{de}$ | 0.13 ± 0.02 |
| S12 | 1.84 ± 0.33$^c$ | 0.30 ± 0.05$^e$ | 0.46 ± 0.03$^d$ | 1.19 ± 0.20$^{de}$ | 0.41 ± 0.06$^c$ | 0.16 ± 0.03$^c$ | 1.82 ± 0.32$^b$ | 1.53 ± 0.04$^{bc}$ | 0.11 ± 0.00$^{de}$ | 0.07 ± 0.00$^b$ | 0.94 ± 0.02$^{bc}$ | 0.09 ± 0.00$^b$ | 0.07 ± 0.00$^c$ | 0.55 ± 0.00$^b$ |
| S13 | 0.14 ± 0.00$^g$ | 0.62 ± 0.02$^{bc}$ | 0.58 ± 0.02$^c$ | 0.43 ± 0.01$^g$ | 1.43 ± 0.04$^a$ | 0.22 ± 0.01$^b$ | 1.05 ± 0.03$^e$ | 0.07 ± 0.00$^i$ | 0.08 ± 0.00$^{ef}$ | 0.05 ± 0.00$^{cd}$ | 0.11 ± 0.00$^{gh}$ | 0.05 ± 0.00$^c$ | 0.04 ± 0.00$^{de}$ | 0.14 ± 0.00$^{fg}$ |
| S14 | 1.72 ± 0.07$^{cd}$ | 0.44 ± 0.03$^d$ | 1.02 ± 0.07$^a$ | 1.77 ± 0.12$^a$ | 1.34 ± 0.07$^b$ | 0.16 ± 0.02$^c$ | 1.72 ± 0.09$^b$ | 0.87 ± 0.15$^d$ | 0.12 ± 0.02$^d$ | 0.15 ± 0.03$^a$ | 0.86 ± 0.14$^{cd}$ | 0.19 ± 0.02$^a$ | 0.05 ± 0.01$^d$ | 0.35 ± 0.06$^d$ |
| S15 | 0.20 ± 0.01$^g$ | 0.06 ± 0.01$^{fg}$ | 0.02 ± 0.00$^h$ | 0.17 ± 0.00$^{gh}$ | 0.07 ± 0.01$^h$ | 0.02 ± 0.00$^f$ | 1.02 ± 0.06$^e$ | 0.73 ± 0.09$^{de}$ | 0.08 ± 0.01$^{fg}$ | 0.02 ± 0.00$^e$ | 0.64 ± 0.08$^e$ | 0.05 ± 0.00$^c$ | 0.03 ± 0.00$^{fgh}$ | 0.17 ± 0.02$^{efg}$ |

F1: quercetin-3-*O*-glucuronide; F2: quercetin-3-*O*-galactoside; F3: isoquercitrin; F4: kaempferol-3-*O*-glucuronide; F5: kaempferol-3-*O*-glucoside; F6: quercetin-3-*O*-rhamnoside; F7: kaempferol-3-*O*-rhamnoside; nd: not detected. Values within different superscripts are different in the same column at 0.05 levels.

**Table 3.** The content (mg/g) of individual triterpenoid in ethanol and water extracts of *C. paliurus* leaves.

| Populations | Ethanol | | | | | | Water | | | |
|---|---|---|---|---|---|---|---|---|---|---|
| | T1 | T2 | T3 | T4 | T5 | T6 | T1 | T2 | T5 | T6 |
| S1 | 3.35 ± 0.03$^b$ | 2.00 ± 0.00$^a$ | 0.73 ± 0.01$^{ef}$ | 2.38 ± 0.01$^c$ | 0.86 ± 0.01$^b$ | 0.47 ± 0.00$^c$ | 0.16 ± 0.01$^b$ | 0.17 ± 0.03$^{cd}$ | nd | 0.07 ± 0.01$^a$ |
| S2 | 0.93 ± 0.06$^{ef}$ | 0.21 ± 0.03$^f$ | 0.76 ± 0.02$^{ef}$ | 0.32 ± 0.01$^{de}$ | 0.20 ± 0.03$^{ef}$ | 0.14 ± 0.00$^{fg}$ | 0.09 ± 0.01$^{cd}$ | 0.13 ± 0.00$^e$ | nd | 0.05 ± 0.00$^{cd}$ |
| S3 | 2.87 ± 0.10$^b$ | 1.48 ± 0.04$^b$ | 1.91 ± 0.05$^d$ | 1.88 ± 0.07$^c$ | 0.72 ± 0.02$^{bc}$ | 0.36 ± 0.02$^d$ | 0.16 ± 0.01$^b$ | 0.16 ± 0.00$^{cde}$ | nd | 0.05 ± 0.00$^{cd}$ |
| S4 | 0.26 ± 0.04$^g$ | 0.27 ± 0.01$^f$ | Nd | 0.07 ± 0.00$^e$ | 0.04 ± 0.01$^f$ | 0.05 ± 0.00$^{gh}$ | 0.002 ± 0.0003$^f$ | 0.13 ± 0.01$^e$ | nd | 0.03 ± 0.00$^{fg}$ |
| S5 | 1.87 ± 0.05$^c$ | 1.35 ± 0.04$^b$ | 3.11 ± 0.08$^c$ | 2.11 ± 0.10$^c$ | 0.85 ± 0.07$^{bc}$ | 0.42 ± 0.01$^{cd}$ | 0.24 ± 0.00$^a$ | 0.18 ± 0.00$^{abc}$ | 0.04 ± 0.00$^c$ | 0.04 ± 0.00$^{efg}$ |
| S6 | 1.51 ± 0.03$^{cd}$ | 0.90 ± 0.02$^{cd}$ | 0.31 ± 0.01$^{fg}$ | 0.89 ± 0.02$^d$ | 0.77 ± 0.02$^{bc}$ | 0.71 ± 0.03$^{ab}$ | 0.13 ± 0.00$^{bc}$ | 0.17 ± 0.00$^e$ | nd | 0.04 ± 0.00$^{de}$ |
| S7 | 1.50 ± 0.04$^{cd}$ | 0.63 ± 0.02$^{de}$ | 0.77 ± 0.04$^{ef}$ | 0.79 ± 0.02$^d$ | 0.31 ± 0.02$^{de}$ | 0.22 ± 0.01$^{ef}$ | 0.17 ± 0.03$^b$ | 0.14 ± 0.01$^{de}$ | nd | 0.04 ± 0.00$^{efg}$ |
| S8 | 1.87 ± 0.15$^c$ | 0.91 ± 0.06$^c$ | 5.33 ± 0.39$^b$ | 5.00 ± 0.39$^b$ | 0.69 ± 0.04$^c$ | 0.37 ± 0.03$^d$ | 0.08 ± 0.00$^d$ | 0.21 ± 0.00$^{ab}$ | 0.05 ± 0.00$^b$ | 0.07 ± 0.00$^{ab}$ |
| S9 | 0.60 ± 0.00$^{fg}$ | 0.14 ± 0.02$^f$ | 0.22 ± 0.02$^{fg}$ | 0.23 ± 0.04$^{de}$ | 0.16 ± 0.00$^{ef}$ | 0.02 ± 0.00$^h$ | nd | nd | nd | 0.05 ± 0.00$^{cd}$ |
| S10 | 0.23 ± 0.04$^g$ | 0.28 ± 0.03$^f$ | 0.14 ± 0.02$^{fg}$ | 0.19 ± 0.02$^{de}$ | 0.04 ± 0.01$^f$ | 0.05 ± 0.01$^{gh}$ | 0.08 ± 0.01$^{de}$ | 0.18 ± 0.02$^{abc}$ | 0.03 ± 0.00$^d$ | 0.04 ± 0.01$^{def}$ |
| S11 | 4.60 ± 0.51$^a$ | 2.22 ± 0.23$^a$ | 6.09 ± 0.68$^a$ | 6.36 ± 0.72$^a$ | 1.26 ± 0.14$^a$ | 0.80 ± 0.09$^a$ | nd | 0.13 ± 0.00$^e$ | nd | 0.06 ± 0.00$^{bc}$ |
| S12 | 1.81 ± 0.32$^c$ | 1.61 ± 0.25$^b$ | Nd | 1.77 ± 0.33$^c$ | 0.75 ± 0.13$^{bc}$ | 0.39 ± 0.07$^{cd}$ | 0.16 ± 0.01$^b$ | 0.21 ± 0.01$^a$ | 0.03 ± 0.00$^e$ | 0.03 ± 0.00$^g$ |
| S13 | 0.71 ± 0.02$^{efg}$ | 0.34 ± 0.01$^f$ | 0.57 ± 0.02$^{fg}$ | 0.46 ± 0.01$^{de}$ | 0.83 ± 0.03$^{bc}$ | 0.66 ± 0.02$^b$ | 0.005 ± 0.0008$^f$ | nd | nd | 0.03 ± 0.00$^g$ |
| S14 | 1.13 ± 0.09$^{de}$ | 0.41 ± 0.02$^{ef}$ | 1.23 ± 0.18$^e$ | 0.61 ± 0.05$^{de}$ | 0.24 ± 0.02$^{de}$ | 0.19 ± 0.01$^{ef}$ | 0.04 ± 0.00$^e$ | nd | nd | 0.06 ± 0.00$^{bc}$ |
| S15 | 0.79 ± 0.02$^{ef}$ | 0.26 ± 0.05$^f$ | 0.46 ± 0.03$^{fg}$ | 0.21 ± 0.00$^{de}$ | 0.40 ± 0.04$^d$ | 0.25 ± 0.01$^e$ | 0.21 ± 0.04$^a$ | 0.17 ± 0.01$^{bc}$ | 0.07 ± 0.00$^a$ | 0.04 ± 0.00$^{def}$ |

T1: arjunolic acid; T2: cyclocaric acid B; T3: pterocaryoside B; T4: pterocaryoside A; T5: hederagenin; T6: oleanolic acid; nd: not detected. Values within different superscripts are different in the same column at 0.05 levels.

Bioactive metabolites of *C. paliurus* leaves varied with solvent and population, which corresponded with previous results [13]. Solvent extraction is a commonly used method for obtaining plant extracts. However, the extraction efficiency has a correlation to the polarity of solvents and extracted compounds [29]. The solubility of the compounds can be changed by the interactions between solvents and extracted compounds [30]. Our results were not consistent with the results from Heo et al. [25], where the water extracts of indigo leaves had higher total polyphenol and total flavonoid content than that of the ethanol extracts. Results from this study showed that ethanol solvent was more effective for the extraction of flavonoids and triterpenoids of *C. paliurus* leaves. In addition, no pterocaryoside A and pterocaryoside B were detected in the water extracts, and only minor hederagenin and 4,5-di-*O*-caffeoyluinic acid were observed in fewer populations, which basically agreed with a previous study [13]. These results indicated that the extraction solvents not only affected the contents of bioactive metabolites but also its composition in *C. paliurus* extracts. Moreover, contents of total polyphenol and total flavonoid in both ethanol and water extracts of *C. paliurus* leaves were considerably higher than those of Tossa jute leaves [31]. However, in the water extracts, they were lower than three *Centaurea* species and *Ganoderma adspersum* [32,33]. In the ethanol extracts, they are comparable with pyrola leaves from some locations in Northeast China, but lower than *Ruta chalepensis* and olive leaves [26,34,35].

### 3.2. Effects of Solvent and Geographical Origin on Antioxidant Activity

Solvent and natural population had significant effects on the scavenging capacity of DPPH and ABTS radical and reducing power of the *C. paliurus* extracts, and the significant interaction effects between solvent and population were also observed on antioxidant property (Supplementary Table S2). Ethanol extracts showed stronger antioxidant capacity when compared with water extracts in most *C. paliurus* populations. In the ethanol extracts, the strongest capacity of scavenging DPPH radical was observed in S15 population ($IC_{50}$ = 0.34 mg/mL), while S3 ($IC_{50}$ = 0.64 mg/mL), S6 ($IC_{50}$ = 0.59 mg/mL) and S7 ($IC_{50}$ = 0.50 mg/mL) populations were the best to eliminate ABTS radical. Moreover, the strongest reducing power occurred in the ethanol extracts of S14 population ($EC_{50}$ = 0.71 mg/mL) (Table 4). Simultaneously, in the water extracts, S6 and S7 populations showed better performance of scavenging DPPH and ABTS radical and the strongest reducing power was achieved in S6 and S12 populations (Table 4). Moreover, S9 population had the lowest antioxidant capacity in both ethanol and water extracts of *C. paliurus* leaves (Table 4).

Our results corresponded with previous reports of Liu et al. [20] and Zhou et al. [17] who indicated that the antioxidant activity of *C. paliurus* varied with the geographical locations. The ethanol extracts of *C. paliurus* leaves showed stronger antioxidant capacity than the water extracts, which were in line with the extracts of *F. beecheyana* and *S. chinensis* [23,24]. Thus, the higher yields of bioactive compounds were likely to contribute to the stronger antioxidant property in the ethanol extracts. However, our results were not consistent with the results from Heo et al. [25], where higher contents of total flavonoid and total polyphenol were detected in the water extracts of indigo leaves when compared to the ethanol extracts. Compared with other plant species, the scavenging capacity of DPPH radical and reducing power in the ethanol extracts of *C. paliurus* leaves were weaker than the ethanol extracts of pyrola leaves, but the scavenging capacity of ABTS was similar between the two plants from specific locations [34]. Most populations were higher than goji berry and *Aspalathus Linearis* and some populations were comparable to *R. tomentosa* in terms of the scavenging capacity of DPPH radical [36–38]. The scavenging capacity of ABTS of some populations was higher than specific goji berry and Solanaceae species [36,39]. However, the antioxidant activity of the water extracts in the present study was weaker in comparison with a previous study [17]. It was likely that the extraction method, especially the solid-liquid ratio, affected the antioxidant activity, as the solid-liquid ratio was positively correlated with the yield of bioactive compounds and they might contribute to higher antioxidant activity [30].

**Table 4.** The antioxidant activity of ethanol and water extracts of *C. paliurus* leaves.

| Populations | Ethanol | | | Water | | |
|---|---|---|---|---|---|---|
| | DPPH | ABTS | Reducing Power | DPPH | ABTS | Reducing Power |
| S1 | $0.52 \pm 0.01^f$ | $0.75 \pm 0.02^{cde}$ | $1.10 \pm 0.01^{fg}$ | $1.24 \pm 0.01^{ef}$ | $1.71 \pm 0.02^{ef}$ | $2.48 \pm 0.05^f$ |
| S2 | $0.98 \pm 0.02^{c-f}$ | $1.42 \pm 0.04^{b-e}$ | $1.03 \pm 0.01^g$ | $2.78 \pm 0.05^d$ | $3.68 \pm 0.20^d$ | $2.57 \pm 0.03^f$ |
| S3 | $0.47 \pm 0.00^f$ | $0.64 \pm 0.01^e$ | $1.00 \pm 0.01^g$ | $1.41 \pm 0.01^{ef}$ | $1.81 \pm 0.02^{ef}$ | $1.95 \pm 0.02^{hi}$ |
| S4 | $1.35 \pm 0.01^{cde}$ | $1.88 \pm 0.03^{bcd}$ | $1.82 \pm 0.02^{bc}$ | $4.44 \pm 0.11^b$ | $4.72 \pm 0.12^c$ | $3.86 \pm 0.00^c$ |
| S5 | $0.81 \pm 0.08^{ef}$ | $1.19 \pm 0.01^{b-e}$ | $1.28 \pm 0.02^{ef}$ | $1.17 \pm 0.04^{ef}$ | $1.55 \pm 0.06^{ef}$ | $1.82 \pm 0.06^i$ |
| S6 | $0.40 \pm 0.02^f$ | $0.59 \pm 0.02^e$ | $0.99 \pm 0.01^g$ | $0.88 \pm 0.00^f$ | $1.18 \pm 0.01^f$ | $1.59 \pm 0.01^j$ |
| S7 | $0.92 \pm 0.01^{def}$ | $0.50 \pm 0.01^e$ | $1.15 \pm 0.01^{fg}$ | $0.90 \pm 0.01^f$ | $1.26 \pm 0.01^f$ | $2.09 \pm 0.10^{gh}$ |
| S8 | $5.87 \pm 0.64^b$ | $6.68 \pm 0.92^a$ | $1.91 \pm 0.02^{bc}$ | $3.65 \pm 0.08^c$ | $5.48 \pm 0.13^b$ | $4.26 \pm 0.04^b$ |
| S9 | $7.23 \pm 0.67^a$ | $7.51 \pm 1.15^a$ | $3.48 \pm 0.16^a$ | $10.65 \pm 0.69^a$ | $11.05 \pm 0.67^a$ | $5.97 \pm 0.11^a$ |
| S10 | $0.67 \pm 0.00^{ef}$ | $0.98 \pm 0.02^{b-e}$ | $1.34 \pm 0.01^e$ | $1.39 \pm 0.02^{ef}$ | $2.04 \pm 0.15^e$ | $2.18 \pm 0.04^g$ |
| S11 | $1.65 \pm 0.07^c$ | $2.03 \pm 0.09^b$ | $1.77 \pm 0.01^c$ | $3.38 \pm 0.11^c$ | $5.11 \pm 0.45^{bc}$ | $3.31 \pm 0.04^d$ |
| S12 | $0.47 \pm 0.01^f$ | $0.74 \pm 0.04^{de}$ | $1.03 \pm 0.09^g$ | $1.35 \pm 0.02^{ef}$ | $1.69 \pm 0.06^{ef}$ | $1.45 \pm 0.02^j$ |
| S13 | $1.53 \pm 0.05^{cd}$ | $1.90 \pm 0.06^{bc}$ | $1.99 \pm 0.15^b$ | $3.93 \pm 0.02^{bc}$ | $3.50 \pm 0.09^d$ | $3.27 \pm 0.12^d$ |
| S14 | $0.50 \pm 0.03^f$ | $0.78 \pm 0.02^{cde}$ | $0.71 \pm 0.02^h$ | $1.47 \pm 0.02^e$ | $2.05 \pm 0.02^e$ | $3.01 \pm 0.06^e$ |
| S15 | $0.34 \pm 0.00^f$ | $1.44 \pm 0.09^{b-e}$ | $1.54 \pm 0.02^d$ | $2.25 \pm 0.03^d$ | $1.22 \pm 0.01^f$ | $2.91 \pm 0.01^e$ |

DPPH: the $IC_{50}$ of DPPH; ABTS: the $IC_{50}$ of ABTS; Reducing power: the $EC_{50}$ of reducing power; $IC_{50}$ (mg/mL): the sample concentration at which the radical was scavenged by 50%; $EC_{50}$ (mg/mL): the absorbance was 0.5 for reducing power. Values within different superscripts are different in the same column at 0.05 levels.

## 3.3. Effects of Solvent and Geographical Origin on Anticancer Activity

The antiproliferative effects of *C. paliurus* leaves were investigated against A549, HCT-116, HeLa, HepG2, MCF-7, PANC-1 human cancer cell lines and one normal cell line, HEK-293 by MTT. As displayed in Supplementary Table S2, the solvent and population significantly affected the antiproliferative activity of *C. paliurus* leaves. The *C. paliurus* extracts were able to inhibit the growth of cancer cell lines in a dose-dependent manner. In general, the ethanol extracts exhibited considerable antiproliferation effects when compared with the water extracts (Table 5). However, the antiproliferation effects of *C. paliurus* extracts varied with cancer cell types. The HeLa cell was the most sensitive to the extracts of *C. paliurus* leaves, and the $IC_{50}$ values of the ethanol and water extracts among *C. paliurus* populations varied from 0.13 to 0.42 mg/mL and from 0.15 to 2.24 mg/mL, respectively. The ethanol and water extracts of S6 population showed the best antiproliferative activity on A549 and MCF-7 cells. In the ethanol extracts, S7, S8 and S9 populations exhibited the strongest antiproliferative effects on HepG2 cell ($IC_{50}$ = 0.25 mg/mL), while S7 and S8 populations had the best inhibition effects on the proliferation of PANC-1 cell ($IC_{50}$ = 0.52 mg/mL).

MTT assay as calorimetric method is commonly used for detecting cell viability. To the best of our knowledge, this is the first report to demonstrate the antiproliferation effects of *C. paliurus* leaves on different cancer cell types. The *C. paliurus* leaves showed the property of noncytotoxic activity as the $IC_{50}$ values of all cells were more than 30 μg/mL [40]. Our results were in harmony of the results from Dahham et al. [41], where the ethanol extracts of *Pandanus tectorius* fruits had stronger antiproliferation effects on human cancer cells than the water extracts. Moreover, *C. paliurus* extracts showed stronger antiproliferation effects on HCT-116 ($IC_{50}$ = 0.84 mg/mL), MCF-7 ($IC_{50}$ = 1.5 mg/mL), HeLa ($IC_{50}$ = 1.4 mg/mL) and HepG2 ($IC_{50}$ = 4.1 mg/mL) cancer cells than chlorogenic acid complex isolated from green coffee beans [42]. The antiproliferation effects of the ethanol extracts from most *C. paliurus* populations on A-549, HCT-116, HepG2, MCF-7 cancer cells was weaker than that from the three algal species, but five *C. paliurus* populations showed better antiproliferation effects on HeLa cancer cell than *Laurencia majuscule* species [43]. In terms of antiproliferation effects on HeLa cells, *C. paliurus* can be compared with of Thai medicine plants [44]. The proliferation inhibition of *C. paliurus* leaves on HeLa deserves further research due to the greatest antiproliferation effects among the cancer cells studied in this study. The *C. paliurus* extracts also inhibited the proliferation of normal kidney cell HEK-293, which was in accordance with the results from Liu et al. [45] and Heo et al. [25] who indicated that *Macleaya cordata* and indigo plants had antiproliferation effects on normal human cells (fetal lung fibroblast cell MRC5 and kidney cell HEK-293).

**Table 5.** The antiproliferation effects (IC$_{50}$ values) of ethanol and water extracts of *C. paliurus* leaves on different cell lines.

| Populations | Ethanol | | | | | | | Water | | | | | | |
|---|---|---|---|---|---|---|---|---|---|---|---|---|---|---|
| | A549 | HCT-116 | HEK-293 | HeLa | HepG2 | MCF-7 | PANC-1 | A549 | HCT-116 | HEK-293 | HeLa | HepG2 | MCF-7 | PANC-1 |
| S1 | 0.72 ± 0.03de | 0.66 ± 0.02b | 0.51 ± 0.02b | 0.15 ± 0.01g | 0.41 ± 0.00e | 0.32 ± 0.00def | 0.63 ± 0.00cd | 1.34 ± 0.05cd | 1.04 ± 0.04efg | 8.73 ± 1.17a | 0.68 ± 0.03c | 1.60 ± 0.05ef | 0.72 ± 0.02efg | 1.89 ± 0.09cde |
| S2 | 0.87 ± 0.02b | 0.79 ± 0.00a | 0.40 ± 0.00d | 0.42 ± 0.01a | 0.41 ± 0.00e | 0.33 ± 0.01de | 0.85 ± 0.01a | 3.43 ± 0.52b | 1.71 ± 0.27d | 1.33 ± 0.08bcd | 0.80 ± 0.00bc | 1.52 ± 0.07f | 1.26 ± 0.03bc | 4.82 ± 0.48b |
| S3 | 0.53 ± 0.01j | 0.51 ± 0.00de | 0.33 ± 0.01fg | 0.28 ± 0.03bcd | 0.34 ± 0.01g | 0.38 ± 0.02bc | 0.54 ± 0.02fg | 2.24 ± 0.46bc | 1.01 ± 0.03efg | 1.05 ± 0.07bcd | 0.22 ± 0.01fg | 2.08 ± 0.03def | 0.75 ± 0.02ef | 2.79 ± 0.29c |
| S4 | 0.96 ± 0.03a | 0.80 ± 0.00a | 0.35 ± 0.02ef | 0.22 ± 0.02def | 0.57 ± 0.01a | 0.41 ± 0.01ab | 0.86 ± 0.03a | - | - | 1.04 ± 0.05bcd | 0.38 ± 0.01ef | 2.72 ± 0.14cd | 1.14 ± 0.08cd | - |
| S5 | 0.60 ± 0.01gh | 0.51 ± 0.01def | 0.68 ± 0.02a | 0.13 ± 0.00g | 0.44 ± 0.00d | 0.35 ± 0.01cd | 0.57 ± 0.01efg | - | 1.85 ± 0.14cd | - | 2.10 ± 0.17a | 3.65 ± 0.57c | 1.45 ± 0.05a | 2.70 ± 0.27cd |
| S6 | 0.35 ± 0.01k | 0.48 ± 0.00efg | 0.11 ± 0.01i | 0.22 ± 0.02ef | 0.38 ± 0.01f | 0.22 ± 0.04g | 0.65 ± 0.01bc | 0.54 ± 0.01d | 0.51 ± 0.02g | - | 0.30 ± 0.002efg | 1.03 ± 0.01f | 0.34 ± 0.01h | 1.08 ± 0.02e |
| S7 | 0.64 ± 0.02fg | 0.47 ± 0.01g | 0.20 ± 0.02h | 0.28 ± 0.00b-e | 0.25 ± 0.00k | 0.38 ± 0.01bc | 0.52 ± 0.02g | 0.87 ± 0.04cd | 0.83 ± 0.03efg | 8.22 ± 0.53a | 0.46 ± 0.02de | 1.24 ± 0.00f | 0.60 ± 0.02g | 1.24 ± 0.03e |
| S8 | 0.77 ± 0.01c | 0.52 ± 0.01d | 0.38 ± 0.01de | 0.34 ± 0.03b | 0.25 ± 0.00k | 0.30 ± 0.01ef | 0.52 ± 0.02g | 1.33 ± 0.07cd | 2.56 ± 0.48b | 2.08 ± 0.24b | 0.39 ± 0.03def | 5.21 ± 0.55b | 1.40 ± 0.11ab | 4.86 ± 0.77b |
| S9 | 0.76 ± 0.00cd | 0.49 ± 0.01efg | 0.43 ± 0.01c | 0.31 ± 0.02bc | 0.25 ± 0.00k | 0.38 ± 0.02bc | 0.54 ± 0.01fg | - | - | 1.16 ± 0.06bcd | 0.96 ± 0.02b | 5.21 ± 0.55b | 1.28 ± 0.07bc | - |
| S10 | 0.62 ± 0.01g | 0.53 ± 0.02d | 0.22 ± 0.00h | 0.29 ± 0.02bc | 0.30 ± 0.00i | 0.30 ± 0.01ef | 0.53 ± 0.01fg | 6.84 ± 1.28a | 1.36 ± 0.17de | 1.25 ± 0.09bcd | 0.20 ± 0.00fg | 5.16 ± 0.25b | 1.22 ± 0.05cd | 2.82 ± 0.19c |
| S11 | 0.71 ± 0.01e | 0.48 ± 0.01fg | 0.49 ± 0.02b | 0.27 ± 0.03cde | 0.32 ± 0.01h | 0.44 ± 0.03a | 0.54 ± 0.00fg | - | 2.34 ± 0.15bc | 1.74 ± 0.15bc | 0.68 ± 0.06c | 9.41 ± 0.37a | 1.16 ± 0.03cd | 9.50 ± 0.46a |
| S12 | 0.56 ± 0.00hij | 0.53 ± 0.01d | 0.21 ± 0.01h | 0.17 ± 0.01fg | 0.47 ± 0.01c | 0.40 ± 0.01b | 0.58 ± 0.01def | 0.90 ± 0.02cd | - | 0.87 ± 0.01efg | 0.47 ± 0.01d | 0.15 ± 0.01g | 0.66 ± 0.02fg | 1.44 ± 0.21de |
| S13 | 0.66 ± 0.01f | 0.44 ± 0.00h | 0.36 ± 0.01ef | 0.30 ± 0.05bc | 0.28 ± 0.00j | 0.28 ± 0.01f | 0.55 ± 0.02efg | - | 3.48 ± 0.30a | 0.76 ± 0.01cd | 2.24 ± 0.20a | 5.80 ± 0.95b | - | 9.57 ± 1.04a |
| S14 | 0.55 ± 0.04ij | 0.56 ± 0.01c | 0.31 ± 0.01g | 0.13 ± 0.01g | 0.53 ± 0.01b | 0.29 ± 0.02ef | 0.71 ± 0.03b | 0.97 ± 0.01cd | 0.78 ± 0.01fg | 0.74 ± 0.02cd | 0.60 ± 0.02cd | 1.12 ± 0.02f | 1.08 ± 0.04d | 1.43 ± 0.08e |
| S15 | 0.57 ± 0.01hi | 0.58 ± 0.01c | 0.36 ± 0.01ef | 0.13 ± 0.00g | 0.31 ± 0.00hi | 0.35 ± 0.01cd | 0.60 ± 0.01de | 3.60 ± 0.70b | 1.15 ± 0.04ef | 1.46 ± 0.12bcd | 0.23 ± 0.01fg | 2.68 ± 0.021cde | 0.84 ± 0.03e | 1.10 ± 0.02e |

A549, HCT-116, HEK-293, HeLa, HepG2, MCF-7, PANC-1: the IC$_{50}$ of cells; IC$_{50}$ (mg/mL): the sample concentration at which the growth of cell was inhibited by 50%. Values within different superscripts are different in the same column at 0.05 levels.

### 3.4. Correlation between Phytochemicals and Bioactivity

In order to understand which compounds could be responsible for the antioxidant and antiproliferative activities in *C. paliurus* extracts, the RDA models were performed in the present study. The result indicated that the canonical axes was significant ($p$ = 0.004) and explained 90.0% of the total variation in antioxidant activity (Figure 3A). In the RDA model, the first axes was significant ($p$ = 0.006), and 83.6% of the variation was explained by RDA1 and 5.4% was explained by RDA2 (Figure 3A). Based on the correlation coefficients, total polyphenol (−0.83) showed the greatest correlation with the first axes, followed by total flavonoid (−0.75) and total triterpenoid (−0.62). Apart from 4-*O*-caffeoyluinic acid, kaempferol-3-*O*-glucoside, pterocaryoside A and pterocaryoside B, the explanatory variables were significantly ($p$ < 0.05) associated with antioxidant activity by Monte Carlo permutation test (Table 6). Total polyphenol had the highest explanation of 64.1% and total flavonoid had the second highest explanation of 52.3% (Table 6). Moreover, among the significant variables, phenolic contributed more to antioxidant activity in comparison with triterpenoid with respect to individual compounds (Table 6). Kaempferol-3-*O*-rhamnoside was predominant compounds of antioxidant potential among the individual compounds studied. All the compounds studied were negatively correlated with antioxidant properties except for pterocaryoside A and pterocaryoside B (Figure 3A).

On the other hand, the canonical axes explained 79.1% of the total variation in antiproliferative activity, but the first two axes explained 63.2% and 10.2% of the total variation, respectively (Figure 3B). Based on the correlation coefficients in the RDA model, the first axis was predominantly correlated with total flavonoid (−0.81) and total triterpenoid (−0.77). As indicated in Table 6, the explanatory variables showed significant ($p$ < 0.05) correlation with antiproliferative activity except for 3-*O*-caffeoyluinic acid, 4-*O*-caffeoyluinic acid, kaempferol-3-*O*-glucoside, pterocaryoside A and pterocaryoside B, and total flavonoid and total triterpenoid explained 45.2% and 41.6% of the total variation, respectively. Hederagenin showed the greatest explanation to antiproliferative activity among individual compounds, followed by arjunolic acid, kaempferol-3-*O*-rhamnoside and oleanolic acid. The explanatory variables were positively correlated with the antiproliferation effects on the cancer cells studied except for 4-*O*-caffeoyluinic acid (Figure 3B).

As hydrogen donor, reactive oxygen quenchers, reducing agent and free radical scavenging agent in redox reactions, phenolic compounds are very important for antioxidant activity [38]. Our results showed that phenolics had a greater contribution to antioxidant activity than triterpenoid in the extracts of *C. paliurus* leaves, which confirmed the above viewpoint. Our results were also corresponded with the information given by Zhou et al. [14], where total polyphenol gave the highest explanation for antioxidant property in the extracts of *C. paliurus* leaves. Meanwhile, the total triterpenoid, arjunolic acid, cyclocaric acid B, hederagenin and oleanolic acid also contributed to antioxidant activity, supporting the point that terpenoid compounds were active principles of antioxidant property [46]. In previous studies, Valente et al. [47] and Shaikh et al. [46] reported that the terpenoids of plant extracts could prevent metabolic pathways contributing to cancer, and may be responsible for antiproliferation effects. Gao et al. [48] indicated the antiproliferative activity of oats was closely associated with phenolic compounds. Our results confirmed that terpenoids and phenolic compounds played a vital role in antiproliferation effects in the extracts of *C. paliurus* leaves. Plants possessing abundant phenolics and higher antioxidant capacity often showed higher antiproliferation effects on cancer cells [49]. However, *Pandanus tectorius* showed no cytotoxicity property against HeLa and MCF-7 cell lines even if abundant phenolics and stronger antioxidant properties were observed in its fruits [40]. In the present study, total polyphenol, total flavonoid and total triterpenoid were the top three components contributing to antioxidant and antiproliferative activities. However, no outstanding antiproliferation effects were detected in the ethanol extracts of S7 population, even if the highest total polyphenol, total flavonoid and total triterpenoid and higher scavenging capacity of ABTS radical and reducing power were observed in S7 population. Accordingly, there were other compounds not examined in *C. paliurus* leaves, which might contribute to the antioxidant and antiproliferative effects [50]. For instance, the *C. paliurus* water-soluble polysaccharide exhibited antioxidant and antiproliferation effects on HeLa

cancer cells [15,16]. Besides, the possible synergistic or antagonistic effects between the bioactive metabolites should also be taken into consideration. *C. paliurus* extracts are a complex mixture; their biological activity is attributed to the comprehensive effects of bioactive metabolites. Indeed, the individual compounds would deserve future investigation, as they are likely to show higher bioactivity than the *C. paliurus* extracts.

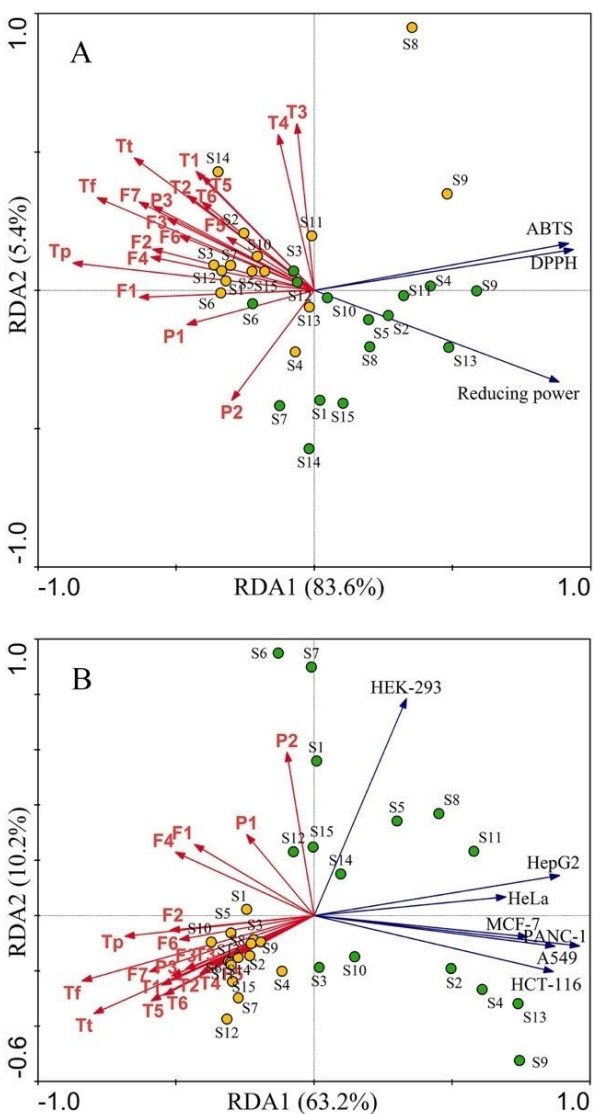

**Figure 3.** Redundancy Analysis (RDA) ordination diagram of bioactive metabolites and antioxidant activity (**A**), bioactive metabolites and antiproliferation effects on cells (**B**). The yellow and green circles represent the ethanol and water extracts of *C. paliurus* population samples, respectively. Abbreviations: Tp: total polyphenol; Tf: total flavonoid; Tt: total triterpenoid; P1: 3-*O*-caffeoyluinic acid; P2: 4-*O*-caffeoyluinic acid; P3: 4,5-di-*O*-caffeoyluinic acid; F1: quercetin-3-*O*-glucuronide; F2: quercetin-3-*O*-galactoside; F3: isoquercitrin; F4: kaempferol-3-*O*-glucuronide; F5: kaempferol-3-*O*-glucoside; F6: quercetin-3-*O*-rhamnoside; F7: kaempferol-3-*O*-rhamnoside; T1: arjunolic acid; T2: cyclocaric acid B; T3: pterocaryoside B; T4: pterocaryoside A; T5: hederagenin; T6: oleanolic acid; DPPH: the $IC_{50}$ of DPPH; ABTS: the $IC_{50}$ of ABTS; Reducing power: the $EC_{50}$ of reducing power; A549, HCT-116, HEK-293, HeLa, HepG2, MCF-7, PANC-1: the $IC_{50}$ of cells.

**Table 6.** The explanation and significance of variables in RDA models.

| Explanatory Variable | Antioxidant | | | Antiproliferation | | |
|---|---|---|---|---|---|---|
| | Variance Explained% | F Value | *p*-Value | Variance Explained% | F | *p*-Value |
| Tp | 64.1% | 49.974 | 0.002 ** | 29.7% | 11.846 | 0.002 ** |
| Tf | 52.3% | 30.705 | 0.002 ** | 45.2% | 23.132 | 0.002 ** |
| Tt | 36.9% | 16.354 | 0.002 ** | 41.6% | 19.908 | 0.002 ** |
| P1 | 17.9% | 6.087 | 0.014 * | 4.9% | 1.445 | 0.216 |
| P2 | 8.3% | 2.544 | 0.096 | 4.4% | 1.303 | 0.248 |
| P3 | 29.1% | 11.503 | 0.004 ** | 17.4% | 5.879 | 0.008 ** |
| F1 | 33.7% | 14.236 | 0.006 ** | 12.6% | 4.043 | 0.028 * |
| F2 | 28.8% | 11.352 | 0.006 ** | 17.2% | 5.828 | 0.014 * |
| F3 | 24.1% | 8.887 | 0.006 ** | 14.1% | 4.601 | 0.022 * |
| F4 | 29.4% | 11.664 | 0.002 ** | 16.5% | 5.535 | 0.012 * |
| F5 | 8.6% | 2.627 | 0.080 | 8.4% | 2.582 | 0.086 |
| F6 | 20.1% | 7.050 | 0.016 * | 15.1% | 4.972 | 0.018 * |
| F7 | 34.2% | 14.554 | 0.002 ** | 22.9% | 8.313 | 0.004 ** |
| T1 | 16.4% | 5.487 | 0.034 * | 19.9% | 6.976 | 0.010 ** |
| T2 | 18.3% | 6.259 | 0.008 ** | 17.3% | 5.847 | 0.014 * |
| T3 | 2.3% | 0.651 | 0.458 | 9.5% | 2.955 | 0.058 |
| T4 | 3.1% | 0.905 | 0.364 | 11.3% | 3.571 | 0.052 |
| T5 | 14.7% | 4.844 | 0.014 * | 23.0% | 8.360 | 0.004 ** |
| T6 | 14.4% | 4.700 | 0.030 ** | 19.4% | 6.749 | 0.006 ** |

Tp: total polyphenol; Tf: total flavonoid; Tt: total triterpenoid; P1: 3-*O*-caffeoyluinic acid; P2: 4-*O*-caffeoyluinic acid; P3: 4,5-di-*O*-caffeoyluinic acid; F1: quercetin-3-*O*-glucuronide; F2: quercetin-3-*O*-galactoside; F3: isoquercitrin; F4: kaempferol-3-*O*-glucuronide; F5: kaempferol-3-*O*-glucoside; F6: quercetin-3-*O*-rhamnoside; F7: kaempferol-3-*O*-rhamnoside; T1: arjunolic acid; T2: cyclocaric acid B; T3: pterocaryoside B; T4: pterocaryoside A; T5: hederagenin; T6: oleanolic acid. * and ** indicated significant at 0.05 and 0.01 levels, respectively.

## 4. Conclusions

In summary, both extraction solvent and geographic origin had significant effects on bioactive metabolites, antioxidant and antiproliferative activities in *C. paliurus* leaves. In most cases, the ethanol solvent was more effective for the extraction of flavonoids and triterpenoids, and higher antioxidant and antiproliferative activities were observed in ethanol extracts. Total polyphenol showed the greatest contribution to the antioxidant activity, while total flavonoid was most responsible for the antiproliferation effects. Moreover, the HeLa cell was the most sensitive to *C. paliurus* extracts among the cancer cell types studied. Further researches should be performed to detect the antioxidant and antitumor effects of individual compounds on HeLa cancer cell, as well as the bioactive mechanisms in *C. paliurus* leaves.

**Supplementary Materials:** The following are available online at http://www.mdpi.com/1999-4907/10/8/625/s1, Figure S1: HPLC chromatograms of the representative sample solution (top) and the responding standard solution containing the 16 quantitative compounds (bottom). 1: 3-*O*-caffeoyluinic acid; 2: 4-*O*-caffeoyluinic acid; 3: quercetin-3-*O*-glucuronide; 4: quercetin-3-*O*-galactoside; 5: isoquercitrin; 6: kaempferol-3-*O*-glucuronide; 7: kaempferol-3-*O*-glucoside; 8: quercetin-3-*O*-rhamnoside; 9: 4,5-di-*O*-caffeoyluinic acid; 10: kaempferol-3-*O*-rhamnoside; 11:arjunolic acid; 12: cyclocaric acid B; 13: pterocaryoside B; 14: pterocaryoside A; 15: hederagenin; 16: oleanolic acid, Table S1: F-values and probability levels from the general linear model (GLM) analysis of bioactive metabolite contents in *C. paliurus* leaves, Table S2: F-values and probability levels from the general linear model (GLM) analysis of antioxidant and anticancer activities in *C. paliurus* leaves.

**Author Contributions:** Conceived and designed the experiment: M.Z. and S.F. Performed the experiment: M.Z., P.C. and Y.L. Analyzed the data: M.Z., P.C. and Y.L. Conceived the paper, wrote the first draft and edited the manuscript: M.Z. and S.F. Supervised the manuscript: X.S.

**Funding:** This work was financially supported by the Jiangsu Province Science Foundation for Youths (No. BK20160926), National Natural Science Foundation of China (No. 31470637), the Priority Academic Program Development of Jiangsu Higher Education Institutions (PAPD) and Doctorate Fellowship Foundation of Nanjing Forestry University. The funders had no role in study design, data collection and analysis, decision to publish, or preparation of the manuscript.

**Acknowledgments:** We acknowledge Xiangxiang Fu, Wanxia Yang, Yang Liu, Bo Deng and Yanni Cao, and Qingliang Liu from Nanjing Forestry University for field and laboratory assistance. We would like to thank Zhiqi Yin from Department of Natural Medicinal Chemistry & State Key Laboratory of Natural Medicines, China Pharmaceutical University for her thoughtful comments to the manuscript.

**Conflicts of Interest:** The authors declare that there are no conflict of interest.

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
