# Peer review of "A Comprehensive Assessment of Bioactive Metabolites, Antioxidant and Antiproliferative Activities of Cyclocarya paliurus (Batal.) Iljinskaja Leaves"

_forests, doi:10.3390/f10080625_

Round 1

Reviewer 1 Report

The manuscript entitled:’A comprehensive assessment of bioactive metabolites, antioxidant and antiproliferative  activities of Cyclocarya paliurus leaves’ is focused on determination of secondary metabolites and bioactivity of extracts prepared from the trees’ leaves from 15 different areas of China. The article is well prepared and brings new information about potential natural medications. Despite I found another articles based on the Cyclocarya paliurus analysis, the studies presented by the Authors are well prepared and are analysed in detail. In my opinion the manuscript is suitable for publication in Forests journal but some points should be improved:

1.      Abstract, line 19: the name of  2,2'-Diphenyl-1-picrylhydrazyl should be without big letter:  2,2'-diphenyl-1-picrylhydrazyl

2.      In my opinion, the IC50 values do not indicate on high antioxidant activity of examined extracts. I think that authors should compare the results with IC50 values for other plants and present the results in more convincing way

3.      Why some of words or sentences are underlined? (ex. lines 84-86)

4.      in vivo should be in vivo (italica)

5.      The HPLC results are well presented in table but personally I think that the chromatogram should be include because some readers would like to check the peaks disjunction. Please provide the chromatogram

6.      Antioxidants studies: Authors provide only IC50 value. I know this is significant for determination antioxidant activity nevertheless in my opinion (and a lot of other scientists) full consideration about the activity is possible when we can observe the changes in free radical scavenging. Namely the studies should be conduct by determination (i.e. dpph) free radical scavenging from reaction initiation up to plateau what results in determination scavenging and this proceed facilitates comparison of antioxidant activity of evaluated compounds/extracts.

Author Response

Dear Reviewer,

Thanks you very much for the comments and corrections for this manuscript. The point-by-point response to your comments are listed as follows:

Point 1: Abstract, line 19: the name of  2,2'-Diphenyl-1-picrylhydrazyl should be without big letter:  2,2'-diphenyl-1-picrylhydrazyl.

Response 1: Yes, we have corrected it. Thank you very much.

Point 2: In my opinion, the IC50 values do not indicate on high antioxidant activity of examined extracts. I think that authors should compare the results with IC50 values for other plants and present the results in more convincing way.

Response 2: Yes, done as you suggested. For the detail, please see the revised version. Thank you for your advice.

Point 3: Why some of words or sentences are underlined? (ex. lines 84-86).

Response 3: We checked the Word file of my original manuscript again (the pending submitted version). We think that the formatting changes of the version cause the mistakes during the process of submitting the manuscript to the submitted system. We have remove the underline formats. Thank you very much.

Point 4: in vivo should be in vivo (italica)

Response 4: Yes, we have corrected it. Thank you very much.

Point 5: The HPLC results are well presented in table but personally I think that the chromatogram should be include because some readers would like to check the peaks disjunction. Please provide the chromatogram.

Response 5: Yes, we have provided the chromatogram in supplementary files (S. Figure 1). Additionally, the sentence “ The chromatograms of the representative sample and the 16 mixed standards were shown in S. Fig.1” was written in my original manuscript (lines 133-134). Thank you very much.

Point 6: Antioxidants studies: Authors provide only IC50 value. I know this is significant for determination antioxidant activity nevertheless in my opinion (and a lot of other scientists) full consideration about the activity is possible when we can observe the changes in free radical scavenging. Namely the studies should be conduct by determination (i.e. dpph) free radical scavenging from reaction initiation up to plateau what results in determination scavenging and this proceed facilitates comparison of antioxidant activity of evaluated compounds/extracts.

Response 6: Yes, six concentration gradients of the extracts were designed by equal interval to detect the free radical scavenging capacity (DPPH and ABTS) in our study. For each sample, the six values of the percent inhibition of radical were obtained and then we should have made a dose-dependent cure graph. However, a large number of samples were tested in the study. For the concise expression and comparison of antioxidant activity of evaluated extracts, we calculated the IC50 value of each sample by the six values. Thank you very much.

Reviewer 2 Report

In this study, Zhou et al., investigated the contents of different metabolites in Cyclocarya leaves and their bioactivities. Although bioactive compounds and their bioactivities in Cyclocarya leaves were reported before, there are still some novelty in this study, including multiple locations of the plants, comparison of two solvent for compound extraction and evaluation of the cause to different activities. I suggest this paper to be accepted until the following comments to be addressed by the authors:

1 The language needs to be revised.

2 From previous studies, polysaccharide is one of the most important bioactive compounds in Cyclocarya leaves. However, polysaccharide is not described enough in the introduction and also is not investigated in this study. Please give sufficient description of polysaccharide in the introduction and explain why it is not contained in this study.

3 In line 72, it is stated that “no information available on the relationship between antioxidant activity and triterpenoid in Cyclocarya extracts”. However, this should be not true. The number 44 reference described this kind of information. Please revise the statement.

4 Please explain why the authors chose ethanol and water for extraction but not other solvent.

5 As shown in Figure 2, the total contents of different metabolites in population S7 are more than other populations. However, the combination of individual compounds in population S7 is not the most one as shown in Tables 1-3. Is this due to the amount of some other compounds which are not measured here are more in population S7. This should be explained.

Author Response

Dear Reviewer,

Thank you very much for taking time to review our manuscript. Based on your comments.we have revised our manuscript, and the point-by-point responses are as follows:

Point 1: The language needs to be revised.

Response 1: Done as you suggested. We have read the whole manuscript in detail, and both some language errors and formats in the text and tables are corrected. Thank you very much.

Point 2: From previous studies, polysaccharide is one of the most important bioactive compounds in Cyclocarya leaves. However, polysaccharide is not described enough in the introduction and also is not investigated in this study. Please give sufficient description of polysaccharide in the introduction and explain why it is not contained in this study.

Response 2: Yes, polysaccharide is very important. Based on your advice, we added the description of polysaccharide in the introduction. For the detail, please see the revised version.

As we all know, polysaccharide is water-soluble and not ethanol-soluble. A previous study confirmed that no polysaccharide was found in the ethanol extracts of Cyclocarya paliurus leaves (Liu, Y.; Cao, Y.N.; Fang, S.Z.; Wang, T.L.; Yin, Z.Q.; Shang, X.L.; Yang, W.X.; Fu, X.X. Antidiabetic effects of Cyclocarya paliurus leaves depends on the contents of antihyperglycemic flavonoids and antihyperlipidemic triterpenoids. Molecules. 2018, 1042, 2-17). Anyway, both ethanol and water extracts were included in our study, so we should pay attention to the polysaccharide of the water extracts. However, the polysaccharide content and antioxidant activity of Cyclocarya paliurus leaves from different geographical locations have been reported in a previous study (Liu, Y.; Fang, S.Z.; Zhou, M.M.; Shang, X.L.; Yang, W.X.; Fu, X.X. Geographic variation in water-soluble polysaccharide content and antioxidant activities of Cyclocarya paliurus leaves. Ind. Crop. Prod. 2018, 121, 180-186). On the other hand, most of anticancer compounds belong to plant secondary metabolites, such as taxol, camptothecin, vinblastine and vincristine. Moreover, many literatures reported that anticancer activity is associated with antioxidant activity. Therefore, we mainly focus on the secondary metabolites of Cyclocarya paliurus leaves and try to seek for key bioactive compounds for further investigation of individual compounds. For the reasons presented above, polysaccharide is not included in this study. Thank you very much.

Point 3: In line 72, it is stated that “no information available on the relationship between antioxidant activity and triterpenoid in Cyclocarya extracts”. However, this should be not true. The number 44 reference described this kind of information. Please revise the statement.

Response 3: Yes, you are right. The number 44 reference had investigated the antioxidant activity of individual compounds of Cyclocarya paliurus triterpenoids but not extracts. All isolated compounds were used to detect their antioxidant effects on FFA-induced hepatic steatosis in HepG2 cells in the study. In order to avoid misunderstandings, we have revised the statement as “less information available”. Thank you very much.

Point 4: Please explain why the authors chose ethanol and water for extraction but not other solvent.

Response 4:  Cyclocarya paliurus phytochemicals were commonly extracted with ethanol and water in previous studies. Based on the results of previous extraction methods studied, 70% ethanol was selected as the extraction solvent because the peak numbers and peak areas reached the highest values in the chromatogram (Cao, Y.N.; Fang, S.Z.; Yin, Z.Q.; Fu, X.X.; Shang, X.L.; Yang, W.X.; Yang, H.M. Chemical fingerprint and multicomponent quantitative analysis for the quality evaluation of Cyclocarya paliurus Leaves by HPLC-Q-TOF-MS. Molecules. 2017, 22). On the other hand, the main utilization of C. paliurus leaves is to make nutraceutical tea for drinking recently, therefore it is very important to investigated the contents and bioactivities of various phytochemicals in the water extracts. Moreover, ethanol and water solvents were characterized by safety, no toxicity and low cost. Thank you very much.

The related references are as follows:

1) Xie, J.H.; Dong C.J.; Nie, S.P.; Li, F.; Wang, Z.J.; Shen, M.Y.; Xie, M.Y. Extraction, chemical composition and antioxidant activity of flavonoids from Cyclocarya paliurus (Batal.) Iljinskaja leaves. Food Chem. 2015, 186, 97-105.

2) Yang, H.M.; Yin, Z.Q.; Zhao, M.G.; Jiang, C.H.; Zhang, J.; Pan, K. Pentacyclic triterpenoids from Cyclocarya paliurus and their antioxidant activities in FFA-induced HepG2 steatosis cells. Phytochemistry. 2018, 151, 119-127.

3) Wu, Z.F.; Meng, F.C.; Cao, L.J.; Jiang, C.H., Zhao, M.G.; Shang, X.L.; Fang, S.Z.; Ye, W.C.; Zhang, Q.W.; Zhang, J.; Yin, Z.Q. Triterpenoids from Cyclocarya paliurus and their inhibitory effect on the secretion of apoliprotein B48 in Caco-2 cells. Phytochemistry. 2017, 142, 76-84.

4) Xie, J.H.; Liu, X.; Shen, M. Y.; Nie, S.P.; Zhang, H.; Li, C.; Gong, D.M.; Xie, M.Y. Purification, physicochemical characterisation and anticancer activity of a polysaccharide from Cyclocarya paliurus leaves. Food Chem. 2013, 136,1453-1460.

5) Xie, J.H.; Xie, M.Y.; Nie, S.P.; Shen, M.Y.; Wang, Y.X.; Li, C. Isolation, chemical composition and antioxidant activities of a water-soluble polysaccharide from Cyclocarya paliurus (Batal.) Iljinskaja. Food Chem. 2010, 119, 1626-1632.

6) Liu, Y.; Cao, Y.N.; Fang, S.Z.; Wang, T.L.; Yin, Z.Q.; Shang, X.L.; Yang, W.X.; Fu, X.X. Antidiabetic effects of Cyclocarya paliurus leaves depends on the contents of antihyperglycemic flavonoids and antihyperlipidemic triterpenoids. Molecules. 2018, 1042.

7) Cao, Y.N.; Fang, S.Z.; Yin, Z.Q.; Fu, X.X.; Shang, X.L.; Yang, W.X.; Yang, H.M. Chemical fingerprint and multicomponent quantitative analysis for the quality evaluation of Cyclocarya paliurus Leaves by HPLC-Q-TOF-MS. Molecules. 2017, 22.

Point 5: As shown in Figure 2, the total contents of different metabolites in population S7 are more than other populations. However, the combination of individual compounds in population S7 is not the most one as shown in Tables 1-3. Is this due to the amount of some other compounds which are not measured here are more in population S7. This should be explained.

Response 5: Yes, it is very likely that these undetected compounds are more in population S7. The total contents of different metabolites were measured by the colorimetric methods, while individual compounds were measured by the HPLC method. In this study, apart from these individual compounds studied, cyclocarioside J, cyclocarioside III and cyclocarioside II belonging to triterpenoids and Kaempferol-3-(6”-(Z)-cinnamylglucoside) belonging to flavonoids were detected but not calculated due to the absence of standards (Cao, Y.N.; Fang, S.Z.; Yin, Z.Q.; Fu, X.X.; Shang, X.L.; Yang, W.X.; Yang, H.M. Chemical fingerprint and multicomponent quantitative analysis for the quality evaluation of Cyclocarya paliurus Leaves by HPLC-Q-TOF-MS. Molecules. 2017, 22, 19-27). Additionally, based on previous studies, some other individual compounds of flavonoids, triterpenoids and phenolics are not identified and measured in this study (see related references as follows). Thank you very much.

 The related references are as follows:

1) Wu, Y.; Li, Y.Y.; Wu, X.; Gao, Z.Z.; Liu, C.; Zhu, M.; Song, Y.; Wang, D.Y.; Liu,J.G.;  Hu, Y.L. Chemical constituents from Cyclocarya paliurus (Batal.) Iljinsk. Biochem. Syst. Ecol. 2014, 216-220.

2) Xie, J.H.; Dong C.J.; Nie, S.P.; Li, F.; Wang, Z.J.; Shen, M.Y.; Xie, M.Y. Extraction, chemical composition and antioxidant activity of flavonoids from Cyclocarya paliurus (Batal.) Iljinskaja leaves. Food Chem. 2015, 186, 97-105.

3) Yang, H.M.; Yin, Z.Q.; Zhao, M.G.; Jiang, C.H.; Zhang, J.; Pan, K. Pentacyclic triterpenoids from Cyclocarya paliurus and their antioxidant activities in FFA-induced HepG2 steatosis cells. Phytochemistry. 2018, 151, 119-127.